# Multi-Level Molecular Differentiation of Populations of the Strand’s Birch Mouse *Sicista strandi* (Rodentia, Dipodoidea)

**DOI:** 10.3390/ani15172605

**Published:** 2025-09-05

**Authors:** Alexey S. Bogdanov, Daria N. Rozhkova, Lyudmila A. Khlyap, Marina I. Baskevich

**Affiliations:** 1Koltzov Institute of Developmental Biology, Russian Academy of Sciences, 119334 Moscow, Russia; darroznature@gmail.com; 2Severtsov Institute of Ecology and Evolution, Russian Academy of Sciences, 119071 Moscow, Russia; khlyap@mail.ru (L.A.K.); mbaskevich@mail.ru (M.I.B.)

**Keywords:** *Sicista strandi*, intraspecific differentiation, molecular variability, the *cytb* gene, the *IRBP* gene

## Abstract

**Simple Summary:**

As previously shown, the Strand’s birch mouse (*Sicista strandi*) is differentiated into two groups, with differences between them reaching the level of species. To clarify the differentiation pattern and structure of *S. strandi*, we analyzed the variability of one mitochondrial gene totally and the fragment of one nuclear gene using additional material. Our results confirmed *S. strandi* division into two main genetically distinct forms: the northern form includes Strand’s birch mice from Belgorod and Kursk regions, while the southern form is represented by specimens from the Greater Caucasus as well as Rostov, Saratov, and Lugansk regions. Within the southern form, the separation of Rostov individuals from other samples was first revealed by the mitochondrial gene. One *S. strandi* specimen from Belgorod region appeared to be the most similar to Strand’s birch mice from Rostov region according to the mitochondrial gene analysis that may indicate existence of a hybrid zone between the northern form and the ‘Don’ line of the southern form. Complex pattern of genetic differentiation traced in *S. strandi* on several levels is of interest for understanding mechanisms of divergence, diversification, and microevolution.

**Abstract:**

Previous molecular studies have demonstrated significant differentiation of the Strand’s birch mouse (*Sicista strandi*) into two groups; differences between them are comparable with those observed between species. However, these studies were based on different, not numerous samples and the small number of individuals. To clarify the differentiation pattern and structure of the Strand’s birch mouse, we analyzed variability of the total mitochondrial *cytb* gene and the nuclear *IRBP* gene fragment using additional material on this species. Our results confirmed its division into two main genetically distinct forms: the northern form includes Strand’s birch mice from Belgorod and Kursk regions, while the southern form incorporates specimens from the Greater Caucasus as well as Rostov, Saratov, and Lugansk regions. In turn, within the southern form, moderate differences were first revealed by the *cytb* gene analysis between individuals from Rostov region and samples from the Greater Caucasus and Saratov region. One specimen from Belgorod region had the *IRBP* haplotypes, typical of this population, but an ‘alien’ mitotype, close to those of Strand’s birch mice from Rostov region. This result may indicate existence of a modern or ancient hybrid zone between the northern form and the ‘Don’ line of the southern form.

## 1. Introduction

According to modern views, the birch mouse genus (*Sicista* Griffith, 1827) comprises 13–17 species [1,2], but the classification is still being revised using genetic approaches, which have the potential to further expand the specific composition of this genus. Thus, significant genetic heterogeneity, which is of great interest for understanding microevolutionary mechanisms, has recently been discovered within some birch mouse species [3], including the Strand’s birch mouse (*Sicista strandi* Formosov, 1931).

The Strand’s birch mouse was initially considered as a form or a subspecies of the northern birch mouse (*S. betulina* Pallas, 1778 s. lato). Later, based on karyotypic peculiarities (2n = 44 in the Strand’s birch mouse versus 2n = 32 in *S. betulina* s. str.), it was proposed that the former should be considered as a separate species [4]. Furthermore, distinctions in the structure of the male genitalia [4], cranial dimensions [5], and nucleotide sequences of several mitochondrial and nuclear genes [3,6,7,8,9,10] have been identified between Strand’s and northern birch mice. As a result, the Strand’s birch mouse is recognized as a sibling species of *S. betulina* s. str., and both of them are included in the betulina group [1,2,11,12,13,14,15,16].

The *S. strandi* range extends from the southern part of the Central Russian Upland (near Kursk) and the Volga Upland to the Greater Caucasus. In a longitudinal direction, it runs from Lugansk region and the eastern coast of Azov Sea to Volga River and the Terek River delta [9,14,16]. The northern and eastern boundaries of the *S. strandi* range are not distinctly defined and require clarification. Distribution of this species appears to be patchy, covering small, isolated groves, forests in ravines, floodplains, and sandy terraces, as well as some steppe areas and protective forest belts; in the Greater Caucasus, Strand’s birch mice inhabit forest-meadows, forest-steppes, and subalpine altitudinal belt [9]. Studies of the Strand’s birch mouse are significantly complicated by sporadicity of its settlements and a low number of individuals in them [14].

Despite its relatively narrow range, *S. strandi* is characterized by clear differentiation by some traits. Two forms can be distinguished within this species, as revealed by the analysis of variability of several mitochondrial and nuclear genes. One of these forms was found in Belgorod region and another in Rostov, Saratov, and Lugansk regions [3,7]. These forms were named as ‘northern’ and ‘southern’, respectively. The genetic differences between them were comparable with interspecific ones. Similar results were obtained in the analysis of the nuclear *IRBP* gene fragment (903 bp), but using material from other populations [8]. According to these data, Strand’s birch mice from Kursk region differed from *S. strandi* specimens from Kabardino-Balkaria and Lugansk region by six fixed nucleotide substitutions that is a high value for a nuclear gene. It is interesting that preliminary studies have also demonstrated some peculiarities in the structure of glans penis and baculum in *S. strandi* males from Kursk region and North Ossetia [4,17]. Furthermore, some low differences were revealed in heterochromatic chromosomal regions and craniometric characteristics for *S. strandi* specimens from Kursk and Saratov regions, on the one hand, and from the Greater Caucasus, on the other hand [5].

However, previous molecular studies of *S. strandi* were based on different samples and markers as well as on the small number of individuals (five specimens maximum) that impedes analysis of these data. The use of such limited material seriously complicates both comparison of variability patterns of different traits (even molecular markers aside) and comprehension of differentiation and the structure of *S. strandi*. Taking into account scarce and fragmentary information about the Strand’s birch mouse, this species may be determined as poorly studied for now. Moreover, possible existence of two sibling species instead of *S. strandi* s. lato is of great interest. The present study aimed to analyze variability of the complete mitochondrial cytochrome *b* gene (*cytb*) and a fragment of the first exon of the nuclear interphotoreceptor retinoid binding protein gene (*IRBP*) in Strand’s birch mice, using all available material, both our own and that from the GenBank database. The *cytb* and *IRBP* genes, as ones of most frequently studied in different species of birch mice [3,6,7,8,18,19] and other rodents, allow to utilize the largest amount of comparative data.

## 2. Materials and Methods

### 2.1. Sample Collection

Our material, which included along with Strand’s birch mice also specimens of *S. subtilis*, *S. severtzovi*, and *S. betulina* as an outgroup, is presented in Table 1. In total, we studied for the first time 22 birch mice of several species for the *cytb* gene and 14 *S. strandi* individuals for the *IRBP* gene. Animals were treated according to conventional international protocols according to the Guidelines for Humane Endpoints for Animals Used in Biomedical Research. Every possible care was taken to reduce the animal’s suffering during capturing and sampling. All the experimental protocols were approved by the Ethics Committee for Animal Research of Koltzov Institute of Developmental Biology, Russian Academy of Sciences in accordance with the Regulations for Laboratory Practice in the Russian Federation (the final protocols are № 37 of 25 June 2020 and № 70 of 25 May 2023).

Along with our material, we analyzed also several sequences of the *cytb* and *IRBP* genes, available in GenBank: five *cytb* gene sequences from *S. strandi* specimens as well as 13 *IRBP* gene sequences from nine *S. strandi*, two *S. betulina*, one *S. subtilis*, and one *S. severtzovi* individuals. Their GenBank accession numbers are listed in Table 1.

Localities, where the animals were caught (including material from GenBank), are also shown in Figure 1.

### 2.2. DNA Extraction, Polymerase Chain Reaction, and Sequencing

Total DNA was extracted from liver, kidney or heart samples fixed in alcohol after treatment of them by proteinase K, deproteinization by phenol-chloroform mixture and precipitation by isopropanol [20]. Besides this, DNA from a part of samples was obtained using the reagent kit ‘DNA-Extran-2’ (Syntol, Moscow, Russia) according to manufacturer’s protocol. Primers used for polymerase chain reaction (PCR) and sequencing of the total mitochondrial *cytb* gene (1140 bp) are listed in Table 2. For this gene amplification, PCR was conducted in a TERCYC thermal cycler (DNA-Technology, Moscow, Russia) in a mixture, containing 25 ng of DNA, 2 μL of 10× Taq buffer, 1.6 μL of 2.5 mM dNTP (Sileks, Moscow, Russia), 4 pM of each primer, 1 unit of Taq polymerase (Syntol, Moscow, Russia), and deionized water to a final volume of 20 μL. PCR conditions were as follows: preheating at 94 °C (3 min) and then 35 cycles, which include 30 s at 94 °C, 1 min at 55 °С, and 1 min at 72 °C; finally, extension of the PCR products was performed at 72 °C (6 min). PCR of the first exon fragment (903 bp) of the nuclear *IRBP* gene was performed in the same way as described earlier [8]. Three forward primers, IRBP-F (5′-AGCAGGCCATGAAGAGTCG-3′), IRBP-F1int (5′-AGCAGCTCATGGGCACTT-3′), and IRBP-F3int (5′-CATTGTGGTGGGTGAGCGGACTG-3′), as well as two reverse primers, IRBP-R (5′-TCATTATCACGGAGGCATCAGC-3′) and IRBP-Rint (5′-CAGATCTCCGTGGTGGTATT-3′), were used. PCR conditions were similar with those for amplification of the *cytb* gene, except for the annealing temperature (57 °C instead of 55 °C). Automatic sequencing was carried out using the ABI PRISM^®^ BigDye^TM^ Terminator v. 3.1 Kit (Applied Biosystems, Foster City, CA, USA) or the NovaDye Terminator Cycle Sequencing Kit (GeneQuest, Moscow, Russia) in the AB 3500 genetic analyzer (Applied Biosystems, Foster City, CA, USA).

Sequences obtained by us in this study have been deposited in GenBank under accession numbers PV848055–PV848076 for the *cytb* gene and PV848077–PV848090 for the *IRBP* gene (Table 1).

### 2.3. Evolutionary Analysis

Dendrograms were built by the Maximum Likelihood (ML) method using IQTree software, version 2.0-rc2 [22,23]. The ModelFinder option [24] was applied to determine optimal model evaluation of nucleotide substitutions for each gene. Standard non-parametric bootstrapping was conducted throughout 1000 pseudo replications. Visualization and processing of phylogenetic trees were performed in FigTree software, version 1.4.4 [25].

The uncorrected mean pairwise genetic *p*-distances (*D*) between species and intraspecific groups were calculated using Mega X software, version 10.1.7 [26].

## 3. Results

### 3.1. Cytb Gene Variability

Good quality chromatograms were obtained using Sic-cytbF and Sic-cytbR primers for Strand’s birch mice from Kursk region as well as for *S. subtilis* and *S. severtzovi* specimens. However, the results of the sequencing, which was performed with the same primers for Strand’s birch mice from the Greater Caucasus, the Don delta, and Saratov region, as well as for *S. betulina* individuals, were ambiguous due to presence of some unresolved sites, demonstrating double peaks in the chromatograms. This can indicate existence of pseudogenes in the genomes of these specimens; similar results were obtained in a previous study [19]. So, to reliably perform the *cytb* gene sequencing in *S. betulina* individuals and Strand’s birch mice from the Greater Caucasus, the Don River delta, and Saratov region, a longer mtDNA fragment was amplified using Sic-cytbF and Sic-DLst primers. This fragment embraces also a part of the control region (*D-loop*) and tRNA genes located between it and the *cytb* gene (Table 2).

As the sequences that we had were of different lengths (1116, 1126, and 1140 bp), we analyzed two sets of them. The first set included the entire *cytb* gene sequences, i.e., all that were obtained in this study and KY967413–KY967414 from the GenBank database. The maximum likelihood dendrogram based on these sequences (Figure 2a) demonstrated distribution of *S. strandi* mitotypes into two significantly separated clades (with a mean genetic distance of *D* = 0.052; see Table 3), which nevertheless had no high statistical support (bootstrap indices of 67% and 99%). One clade (I) was represented by specimens from the Greater Caucasus, Rostov region, and Saratov region, while another clade (II) included individuals from Kursk region. It should be noted that the clade I also appeared to be heterogeneous: Strand’s birch mice from Rostov region (near Don River) formed subclade I-B, which was moderately separated from the most prevalent subclade I-A (bootstrap indices of 98% and 79%, respectively; *D* = 0.023). The latter included Caucasian and Saratov samples.

The second dataset, which was analyzed here, contained all the available *cytb* gene sequences, including incomplete ones from GenBank. The maximum likelihood dendrogram built on the basis of the *cytb* gene fragments (1116 bp) had a topology, similar to that of the previous tree. It demonstrated the same clades and subclades as well as comparable statistical support values and genetic distances (Figure 2b, Table 3). However, the added sample from Belgorod region surprisingly appeared to be heterogeneous. As expected, the mitotypes of two specimens (ZMMU S-181444 and 21) fell into clade II (i.e., together with Strand’s birch mice from Kursk region), but the mitotype of the third individual (1, ZMMU S-181441) was included into subclade I-B, alongside Strand’s birch mice from Rostov region.

### 3.2. IRBP Gene Variability

Among the studied birch mice, five *S. strandi* individuals (№ 03-11, 06-66, 06-69, 06-70, and KF854242) and one *S. subtilis* specimen (№ 03-216) appeared to be heterozygous, but only at one site in all cases. It allows us to separate heterozygous genotypes into haplotypes.

Like the *cytb* gene mitotypes, the *IRBP* gene haplotypes of Strand’s birch mice were distributed in the maximum likelihood dendrogram into two main clades (Figure 3): one clade (I) was represented by specimens from the Greater Caucasus, Saratov, Rostov, and Lugansk regions, while another (II), by all individuals from Kursk and Belgorod regions. These clades had quite high statistical support (bootstrap indices of 96% and 99%, respectively) and demonstrated significant separation (*D* = 0.008), which was comparable with the differences between each of them and *S. betulina* (*D* = 0.007–0.008) or even exceeded the genetic distance between *S. subtilis* and *S. severtzovi* (*D* = 0.002; Table 3). Nevertheless, there are some discrepancies between the dendrograms based on the nuclear and mitochondrial markers. Thus, clades, containing *S. strandi* and *S. betulina* specimens, display trichotomy in the dendrogram built using the *IRBP* gene sequences, possibly due to similar genetic distance values between these groups or insufficient data. Furthermore, this dendrogram does not demonstrate a clear separation of Strand’s birch mice from Rostov region (despite one of them has a unique genotype) and conspecific samples from the Greater Caucasus and Saratov region that was evident in the case of the *cytb* gene. It is also remarkable that one specimen from Belgorod region (1, ZMMU S-181441) demonstrates radically different phylogenetic relationships and positions in the dendrograms (Figure 2b and Figure 3). This individual was clustered by the *IRBP* gene sequences in accordance with its geographical origin, while it appeared to be closest to the representatives of another phylogenetic group, i.e., Strand’s birch mice from Rostov region, by the *cytb* gene.

## 4. Discussion

The analysis of the variability of both the mitochondrial *cytb* gene and the nuclear *IRBP* gene that was conducted in the present study confirms differentiation of *S. strandi* into two main genetically distinct groups. They generally correspond to the northern and southern forms of this species, which were previously described [3,9]. One group incorporates Strand’s birch mice from Belgorod and Kursk regions (the northern form), while another comprises *S. strandi* specimens from the Greater Caucasus, Saratov, and Rostov regions (the southern form). Judging by the *IRBP* gene sequence, the latter group also includes a specimen from Lugansk region. In turn, using additional material, we first discovered moderate subdivision of the southern form based on the *cytb* gene analysis: Strand’s birch mice from Rostov region appeared to be distinct from the Caucasian and Saratov samples. It cannot be excluded that Don River was the main factor, which determined differentiation of the southern form by mitochondrial DNA; in this case, the phylogenetic line revealed in the south of Rostov region could spread northward to the middle reaches of Don. Therefore, at least three levels of differentiation (taking into account also intrapopulation and weak interpopulation variability) can be distinguished in the Strand’s birch mouse by the *cytb* gene. However, unlike the latter, the analysis of the nuclear *IRBP* gene did not demonstrate such distinct separation of Strand’s birch mice from Rostov region from Caucasian and Saratov populations. This may be due to the fact that rates of mutation accumulation and haplotype sorting are lower in nuclear genes than in mitochondrial ones [27,28]. So, the classification of the *S. strandi* specimen from Lugansk region as belonging to any line of the southern form remains uncertain because the individual was not involved to the mitochondrial DNA analyses.

The genetic distance values, which reach 0.05 and more for the *cytb* gene, are usually considered to be at the species level in mammals [29,30]. The distance determined between northern and southern forms of the Strand’s birch mouse for the total *cytb* gene was equal 0.052 (Table 3). Along with the species pair *S. subtilis-S. severtzovi* mentioned above, comparable *p*-distances were earlier revealed between *S. nordmanni* and *S. trizona* (0.068) as well as *S. subtilis* and *S. cimlanica* (0.033) [3]. Thus, significant molecular differences, comparable with interspecific ones in both genes that we studied, as well as absence of ‘intermediate’ mitotypes and the *IRBP* gene haplotypes between northern and southern forms, support the view that they should be considered as distinct species [3,8]. The data on possible presence of the *cytb* pseudogene in the genome of the southern form may also indirectly reflect its obvious diversification from the northern form. Otherwise, the identification of an ‘alien’ mitotype in a single *S. strandi* specimen from Belgorod region may indicate existence of a hybrid zone between the northern form and the ‘Don’ line of the southern form, either currently or in the past. If the hybridization appears to be quite intensive and occurs in various contact zones between the northern and southern forms, they should preferably be considered as intraspecific groups or species in statu nascendi despite their distinct genetic disunity. Another argument in favor of this point of view is narrow, uncharacteristic for most rodent species, distribution area of *S. strandi* northern form, which has so far been found in two districts of Kursk and Belgorod regions only. However, the ranges of this form and others are defined approximately and need clarification; it cannot be ruled out that the northern form actually occupies a much larger area than is currently known. So, a number of problems related to the distribution of intraspecific forms of *S. strandi* and their natural hybridization await their solution in the future, when new data is available.

## 5. Conclusions

Thus, the presented molecular analysis showed complex, multi-level differentiation of *S. strandi*, despite the quite narrow range of this species, and also provided data, which allow us to assume the hybridization of its intraspecific forms. As present and previous studies included material from the main part of the Strand’s birch mouse range, it can be proposed that the structure and differentiation pattern of this species have generally been established. However, the distribution boundaries of all *S. strandi* intraspecific forms and the extent of their reproductive isolation remain unclear, making taxonomic assessments difficult. Further comprehensive research using additional material from a number of new localities, first of all, from the territory of Kursk, Belgorod, Voronezh, Rostov regions and adjacent areas, is needed to study of *S. strandi* intraspecific forms.

## Figures and Tables

**Figure 1 animals-15-02605-f001:**
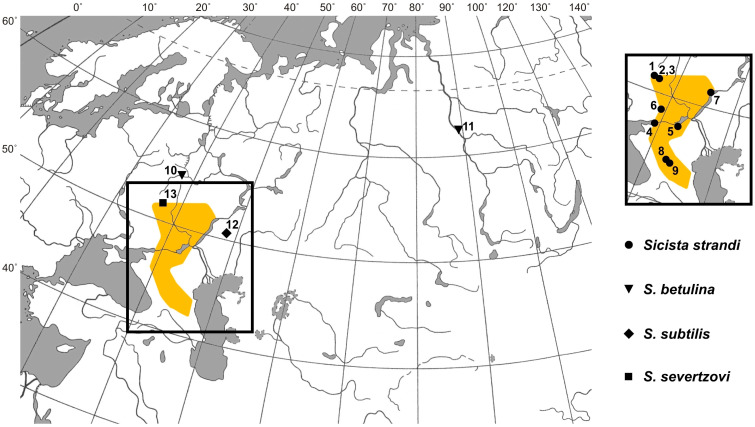
Sampling sites of birch mice of the species *Sicista betulina*, *S. subtilis*, *S. severtzovi* (shown on the large map), and *S. strandi* (shown on a separate map fragment). The numbers of the points are the same as in other figures and tables. Neighboring capture points 2 and 3 are designated by a single symbol. The supposed geographic distribution of *S. strandi* according to our own and published data [9,14,16] is shown in orange shading.

**Figure 2 animals-15-02605-f002:**
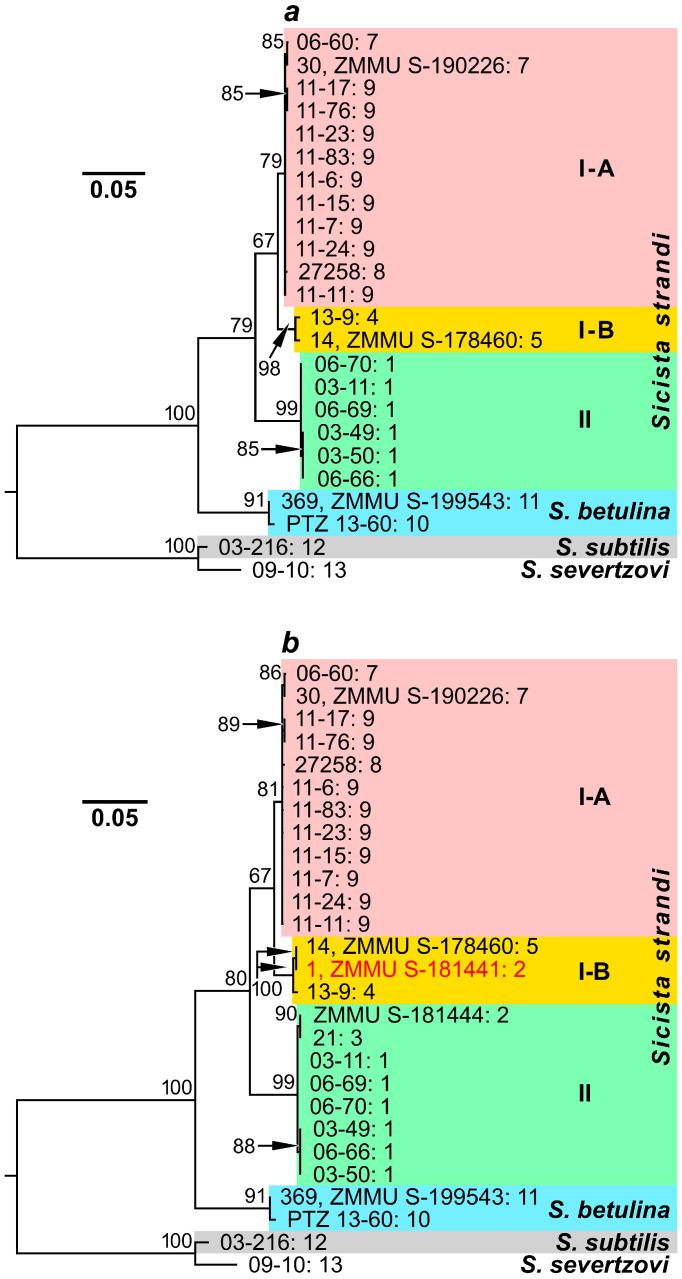
Maximum likelihood dendrograms constructed from a comparison of *S. strandi*, *S. betulina*, *S. subtilis*, and *S. severtzovi* specimens by sequences of (**a**)—the total *cytb* gene (1140 bp); (**b**)—the *cytb* gene fragment (1116 bp). The identification numbers (the collection and/or voucher numbers) of the animals are indicated to the right of the branches before the colon. After the colon, the collection site numbers (see Table 1 and Figure 1) are presented. Bootstrap index values, exceeding 60%, are indicated above the branching nodes of the dendrograms. A specimen, which has various positions in dendrograms built on the basis of mitochondrial and nuclear DNA analyses (see the text below), is marked by red font.

**Figure 3 animals-15-02605-f003:**
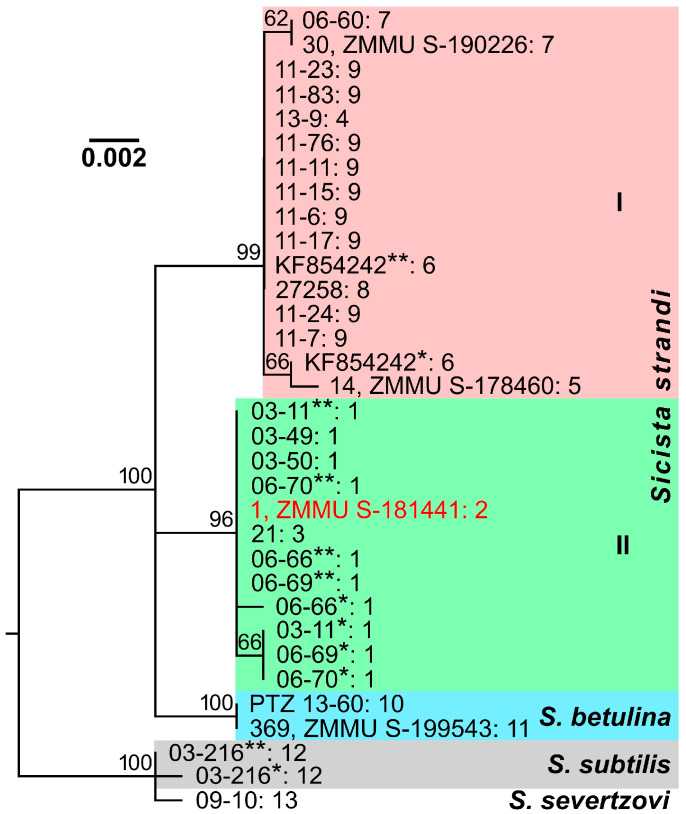
Maximum likelihood dendrogram constructed from a comparison of *S. strandi*, *S. betulina, S. subtilis*, and *S. severtzovi* specimens by sequences of the *IRBP* gene fragment (903 bp). The identification numbers (the collection and/or voucher numbers) or GenBank numbers of the animals are indicated to the right of the branches before the colon. After the colon, the collection site numbers (see Table 1 and Figure 1) are presented. Bootstrap index values, exceeding 60%, are indicated above the branching nodes of the dendrogram. Different haplotypes of the *IRBP* gene, which were derived from heterozygous genotypes of some individuals, are designated by one or two asterisks. A specimen, which has various positions in dendrograms built on the basis of mitochondrial and nuclear DNA analyses (see Figure 2b), is marked by red font.

**Table 1 animals-15-02605-t001:** Samples (our own and derived from GenBank) of several birch mouse species included in this study.

Species	Collection Number	Museum Voucher	Locality and Geographical Coordinates	GenBank Accession Number	
				*cytb*	*IRBP*
*S. strandi*	03-11		1. Russia, Kursk region, vicinities of the city of Kursk (51.58° N, 36.08° E)	**PV848055**	MN175445 ^1^
*S. strandi*	03-49		The same locality	**PV848056**	**PV848077**
*S. strandi*	03-50		The same locality	**PV848057**	**PV848078**
*S. strandi*	06-66		The same locality	**PV848058**	**PV848079**
*S. strandi*	06-69		The same locality	**PV848059**	**PV848080**
*S. strandi*	06-70		The same locality	**PV848060**	MN175446 ^1^
*S. strandi*	1	ZMMU S-181441	2. Russia, Belgorod region, Gubkinsky district, Yamskaya steppe (51.18° N, 37.62° E)	KM397209 ^2, 3, 4^ (1116 bp)	KM397158 ^2, 4^
*S. strandi*	—	ZMMU S-181444	The same locality	MK758092 ^3^ (1126 bp)	—
*S. strandi*	21		3. Russia, Belgorod region, Gubkinsky district	MK259966 ^4^ (1126 bp)	MK259970 ^4^
*S. strandi*	13-9		4. Russia, Rostov region, vicinities of the city of Rostov-on-Don, the Don River delta	**PV848072**	**PV848090**
*S. strandi*	14	ZMMU S-178460	5. Russia, Rostov region, Tsimla sands (47.92° N, 42.50° E)	KY967414 ^3, 4, 5^ (1140 bp)	KY967465 ^4, 5^
*S. strandi*			6. Lugansk region (48.12° N, 39.80° E)	—	KF854242 ^6^
*S. strandi*	30	ZMMU S-190226	7. Russia, Saratov region, Slavianka (51.83° N, 46.25° E)	KY967413 ^3, 4, 5^ (1140 bp)	KY967466 ^4, 5^
*S. strandi*	06-60		The same locality	**PV848061**	**PV848081**
*S. strandi*	27258		8. Russia, Stavropol Krai, 7 km south of the city of Kislovodsk (43.832° N, 42.680° E)	**PV848062**	**PV848082**
*S. strandi*	11-6		9. Russia, the Kabardino-Balkar Republic, Zolsky district, Ekiptsoko (43.68° N, 43.08° E)	**PV848063**	**PV848083**
*S. strandi*	11-7		The same locality	**PV848064**	**PV848084**
*S. strandi*	11-11		The same locality	**PV848065**	**PV848085**
*S. strandi*	11-15		The same locality	**PV848066**	**PV848086**
*S. strandi*	11-17		The same locality	**PV848067**	**PV848087**
*S. strandi*	11-23		The same locality	**PV848068**	MN175447 ^1^
*S. strandi*	11-24		The same locality	**PV848069**	**PV848088**
*S. strandi*	11-76		The same locality	**PV848070**	**PV848089**
*S. strandi*	11-83		The same locality	**PV848071**	MN175448 ^1^
*S. betulina*	PTZ 13-60		10. Russia, Moscow region, Serpukhovsky district (54.915° N, 37.572° E)	**PV848073**	MN175437 ^1^
*S. betulina*	369	ZMMU S-199543	11. Russia, Krasnoyarsk Krai, Turukhansky district, Mirnoye village (62.31° N, 89.02° E)	**PV848074**	MN175444 ^1^
*S. subtilis*	03-216		12. Russia, Saratov region, Alexandrovo-Gaysky district, vicinities of Alexandrov Gay village (50.14° N, 48.57° E)	**PV848075**	MN175450 ^1^
*S. severtzovi*	09-10		13. Russia, Kursk region, vicinities of Barkalovka village (51.56° N, 37.65° E)	**PV848076**	MN175451 ^1^

Note. GenBank accession numbers of sequences first obtained in the present study are shown in bold. GenBank accession numbers from previously published studies are designated by superscripts: ^1^ [8]; ^2^ [18]; ^3^ [7]; ^4^ [3]; ^5^ [6]; ^6^ [19]. Abbreviations: ZMMU—Zoological Museum of Moscow University.

**Table 2 animals-15-02605-t002:** Primers used for PCR conduction and the *cytb* gene sequencing in birch mice.

Species and Populations	Primers and Their Nucleotide Sequences (5′–3′), Source of Data	
	Forward Primers	Reverse Primers
*S. strandi* specimens from Kursk region; *S. subtilis*, *S. severtzovi*	Sic-cytbF (external) (the present study) ACCATCGTTGTCCATTCA	Sic-cytbR (external) (the present study) CCTCATTTTCGGTTTACAAGAC
	Additional internal primers used for sequencing	
	L400-Sic [6] CCATGAGGCCAAATATCATTCTGAGG	MVZ04m [21] GTGGCCCCTCAAAATGATATTTGTCCTC
*S. strandi* specimens from the Greater Caucasus, Rostov and Saratov regions; *S. betulina*	Sic-cytbF (external), see above	Sic-DLst (external) (the present study) AGTGTGCATAGAGAATAAGTCCAG
	Additional internal primers used for sequencing	
	L400-Sic, see above	H670-Sic (the present study) TAGGAATCCTAGGAAGTCTTTG MVZ04m, see above

**Table 3 animals-15-02605-t003:** Average values of pairwise genetic *p*-distances, calculated by comparing sequences of the *cytb* and *IRBP* genes of *S. strandi* intraspecific groups and several other species of birch mice.

Species and Intraspecific Groups	Average Values of Pairwise Genetic *p*-Distances						
	*S. strandi*, Group I-A	*S. strandi*, Group I-B	*S. strandi*, Group I in Total	*S. strandi*, Group II	*S. strandi* in Total	*S. betulina*	*S. subtilis*
*S. strandi*, group I-B	0.023 0.023 —	—	—				
*S. strandi*, group II	0.051 0.052 —	0.061 0.061 —	0.052 0.054 0.008	—	—		
*S. betulina*	0.089 0.089 —	0.090 0.090 —	0.089 0.089 0.008	0.093 0.093 0.007	0.090 0.090 0.008	—	
*S. subtilis*	0.143 0.146 —	0.139 0.141 —	0.143 0.145 0.016	0.146 0.148 0.015	0.144 0.146 0.016	0.150 0.153 0.015	—
*S. severtzovi*	0.141 0.143 —	0.142 0.144 —	0.141 0.143 0.017	0.143 0.146 0.016	0.141 0.144 0.016	0.153 0.155 0.016	0.037 0.038 0.002

Note. In each cell of the table, the upper value of *p*-distance corresponds to the total *cytb* gene (1140 bp), the intermediate value, to the *cytb* gene fragment (1116 bp), and the bottom value, to the *IRBP* gene fragment (903 bp). Designation of the intraspecific groups of *S. strandi* and composition of them are identical to that in ML dendrograms. As the *IRBP* gene does not allow to differentiate groups I-A and I-B of *S. strandi*, the average values of genetic distance for them were not determined.

## Data Availability

Sequences obtained by us in this study have been deposited in GenBank; accession numbers are listed in Table 1.

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
