# Peer review of "Multi-Level Molecular Differentiation of Populations of the Strand’s Birch Mouse Sicista strandi (Rodentia, Dipodoidea)"

_animals, 2025, doi:10.3390/ani15172605_

Round 1

Reviewer 1 Report

Comments and Suggestions for Authors

This study utilized all available materials (including the researchers' own and those from the GenBank database) to conduct variability analyses of the mitochondrial cytb gene and the nuclear IRBP gene. The results confirmed the differentiation of S. strandi into two main genetic clades. The experimental design is rigorous and the data are robust. However, the description in the Methods section requires revision. The manuscript is recommended for revision and publication. Specific suggestions are as follows:

Materials and Methods: 

I apologize, but I was unable to locate the relevant sequences in NCBI (https://www.ncbi.nlm.nih.gov/search/all/?term=PV848055, accessed August 3, 2025) using the accession numbers PV848055–PV848090 listed in Table 1. Please could the authors check whether the accession numbers are correct? Please provide the correct accession numbers and the storage URL/DOI, or the NCBI submission receipt and the expected release date. All sequences mentioned in the manuscript must be made publicly available in some form prior to the paper's publication.

Results: 

Line 189 mentions: "It allows us to separate heterozygous genotypes into haplotypes". It is suggested that the specific phasing algorithm used be detailed in the Methods section. Furthermore, all phased haplotype sequences should be made publicly accessible.

Reviewer 2 Report

Comments and Suggestions for Authors

On the one hand small sample was examined “for the first time 22 birch mice of several species for the cytb gene and 14 S. strandi individuals for the IRBP gene”; on the other hand, only about 600 sequences of Sicista genus are available in GenBank. Authors should clearly present research issues, relevance, not only that questions arise regarding species differentiation. The species under consideration are not protected, IUCN category least concern. So, for example how important are the species under consideration ecologically. In conclusion, at present, I have many questions about the significance of the research and the small sample size, which makes it a very localized study. If the authors can justify the significance of their research, it would be possible to reconsider their manuscript.

L46 about/approximately 13-17 species

L92-93 correct the style of the sentence that avoiding brackets.

In Introduction please broadly describe why cytb and IRBP markers where chosen for this study. Also please provide till the study performed what material for these markers for birch mouse and rodents in general has been accumulated. Lastly please clarify why its is relevant for such kind of studies to choose both mtDNA and nuclear markers, what are benefits of such an approach.

L107 please correct the title and clarify what material is here presented.

L95-97 English of this sentence has to be improved, now it is not clear, what was examined in the present study, and how many samples were taken from GenBank. Are all samples available in the GenBank of the species and genes analyzed in this study, or were only samples from a specific geographical location taken for research?

In methods authors should clearly describe how many sequences of each species were taken from GenBank.

L124 please include sequences of primers

L95-106 in which years authors collected material? It seems that some samples were already analyzed, for instance, IRBP of 06-70 (GenBank acc no MN175446) was obtained previously.

Table S1 it is not clear for me a third figure in some of cells, for instance S. severtzovi  vs. S. subtilis

0.002. Please explain it in table caption.

Please divide Material and Methods into parts, 2.1. and 2.2.

3.1. and 3.2. should be written using every first capitalised letter style

147-149 it should be moved to discussion

L154-155 here is provided that two different set of cytb sequences were obtained, and only in L178 it is cleat the exact length of both sets, so my suggestion to make it clear the length of sequences analyzed at L154-156 rewriting some of sentences

Figure 1 please redo labeling in the figure, instead of collection numbers, authors should use GenBank accession numbers. My additional suggestion to provide in this study generated sequences in bold. Also different localities can be shown using different signs or numbers.

Discussion and Conclusions are well written, but possible hybridization of intraspecific forms of S. strandi is based on few samples. At present, it appears that the results obtained are very preliminary, raising the question of whether it is necessary to publish such interim results or whether it would be preferable to examine additional individuals and then prepare a more robust publication.

Reviewer 3 Report

Comments and Suggestions for Authors

This is interesting and clearly written paper, containing a new original data. I have only two minor notes.

Figure S1. – I don’t understand why this figure is placed in supplementary data file. I would recommend moving this figure to the main text and to use different markers on the map for the two main phylogenetic lineages of Sicista strandi.

Table 2. – In my opinion, this table is not necessary in the main text and can be moved to supplementary data.

Reviewer 4 Report

Comments and Suggestions for Authors

The authors attempted to determine the intraspecific structure of the Strand's birch mouse (Sicista strandi) using mitochondrial and nuclear markers. The results here will further our understanding of the evolutionary history of birch mice Sicista. Below I give a number of suggestions which could help the authors when preparing a revised version of their manuscript.

  1. The Simple Summary and the Abstract are very similar. I recommend shortening the Simple Summary by removing some details. For example, ‘previous molecular studies’, ‘not numerous samples’, ‘moderate differences’ and so on.

Introduction

  1. Lines 60 and 61. ‘The S. strandi range extends from the southern part of the Central Russian Upland (near Kursk) to the Greater Caucasus.’ Judging by the map (Figure 1S), the range extends from the southern part of not only the Central Russian Upland, but also the Volga Upland. Moreover, the northernmost point of S. strandi discovery is located on the Volga Upland.
  2. Line 78. ‘…differ significantly…’ What does ‘significantly’ mean? Based on what criteria? An unfortunate term for molecular genetic differences.

Materials and Methods

  1. Table 1. Locality and geographical coordinates. The authors use different formats to record coordinates, including traditional and decimal formats (DD°MM'; DD.dddddd°). It is necessary to convert all coordinates to a single format, preferably decimal. In locality 8, excessive precision is given — five digits after the decimal point correspond to 1.11 metres.
  2. I recommend that the authors include the sampling sites map (Figure S1) in the main text of the article and move Table 2 (Primers used for...) to the Supplementary materials. Additionally, I recommend colour-coding the sample collection sites on the map according to the clades shown in Figure 1.

Results

  1. I do not see the need to include two dendrograms (based on the complete sequence and fragment of the cytochrome b gene) in Figure 1. The main information is shown in cladogram 1b, which uses a larger number of samples. The difference between the lengths of the sequences used is 24 nucleotide pairs, which does not lead to significant changes in the levels of support for clades and p-distances.
  2. Table S1 should be included in the main body of the article rather than in the supplementary materials. Based on suggestion 5, I propose leaving only the p-distances calculated based on 1116 bp. If the authors had provided the standard error (SE), I believe that the differences between the distances for the complete gene and the fragment would have been insignificant.
  3. Line 201. The term 'star-like structure' is typically employed to describe haplotype networks rather than dendrograms. The position of S. betulina in relation to the two groups of S. strandi is not resolved. In this case, it is preferable to acknowledge the polytomy resulting from insufficient data.

Discussion

  1. In this section, as well as in the Abstract and Introduction sections, the authors note ‘differences, comparable with interspecific’ (lines 27, 75 and 244). However, it should be clarified that this level of similarity was observed only between the sister species S. subtilis and S. severtzovi. Distances between other species of the genus Sicista are usually two to three times higher.
Comments on the Quality of English Language

The article contains spelling and grammatical errors that need to be corrected before publication.

Round 2

Reviewer 2 Report

Comments and Suggestions for Authors

The overall remark 1: please use track changes options in Word or highlight in green or yellow the changes you have made, because it is now very difficult to track down what exactly has been changed in the manuscript

The overall remark 2: In general, I am happy with the progress of manuscript improvement and authors response. However, I invite authors to write several sentences in introduction and in discussion/conclusion highlighting significance of the study as well as limitations of the study. In my opinion still these parts should be improved.

“Comments 1:

Comments and Suggestions for Authors

On the one hand small sample was examined “for the first time 22 birch mice of several species for the cytb gene and 14 S. strandi individuals for the IRBP gene”; on the other hand, only about 600 sequences of Sicista genus are available in GenBank. Authors should clearly present research issues, relevance, not only that questions arise regarding species differentiation. The species under consideration are not protected, IUCN category least concern. So, for example how important are the species under consideration ecologically. In conclusion, at present, I have many questions about the significance of the research and the small sample size, which makes it a very localized study. If the authors can justify the significance of their research, it would be possible to reconsider their manuscript.

Response 1: Of course, S. strandi does not belong to protected species, but taking to account very scarce and fragmentary information about it, this species may be determined as poorly studied at present. So, any new data about S. strandi are actual and important. First of all, genetic data, which indicate possible existence of two sibling species instead S. strandi s. lato, are of great interest. In our opinion, significance of our study is not doubtful and Introduction disclose the problem in sufficient detail.”

Thank you for your reasonable comment. I invite authors to provide two-three sentences why it is signnificant for birch mouse, rodents in general “possible existence of two sibling species instead S. strandi s. lato, are of great interest.“ Please write from a broader perspective, emphasizing and presenting the importance of such research. Also, unofficially speaking, please praise the subject of your research, why it is significant and interesting.  

L113 please indicate genus of organisms analyzed in the tittle

Comments 7: L124 please include sequences of primers

Response 7: For sequencing the IRBP gene fragment were used five primers. All of them were previously published together with PCR conditions (Baskevich et al., 2020). Since we used the same primers and PCR conditions in the present study, duplication of information, which also requires a lot of space, seems unnecessary.” 

Thank you for authors comments, but I have the opposite opinion. The accessibility and reproducibility of scientific results are among the greatest values of science. Personally, I could not easily to find Baskevich et al., 2020 paper. The presentation of primer sequences requires just several lines in paper, and certainly not a lot of space.

L96 change to “Sample Collection and Data Acquisition” or similarly, now the title is not comprehensive and clear

L125 here it is not only Sequencing, please change to “DNA Extraction, PCR and Sequencing” or similarly

L170 please change to “As the sequences that we had were of different lengths (1116 and 1140 bp)...”

Comments 14: Figure 1 please redo labeling in the figure, instead of collection numbers, authors should use GenBank accession numbers. My additional suggestion to provide in this study generated sequences in bold. Also different localities can be shown using different signs or numbers.

Response 14: All sequences generated in this study were initially provided in bold, and different localities were numbered (see Table 1 and note under it as well as legends to all figures). The use of GenBank accession numbers in the dendrograms, in our opinion, is less preferable because it will be more difficult for readers to operate them: GenBank accession numbers are not listed in the table in order as well as they are quite long and similar to each other. So, we gave preference to collection and museum numbers excluding one specimen, which has not any identification numbers and may be distinguished only by GenBank accession number KF854242.” 

Thank you for your comment, I understand that GenBank labelling might be also not very helpful to understand sampling patterns. L186-188 sentence is slightly confusing, it would be wonderful if could improve the sentence highlighting the most important message that after the colon, the collection site numbers (see Table 1 and Figure 187 1) are indicated, other number such as 03-11 are not important for readers to understand figures. Maybe dividing the sentence into two could help.

Comments 15: Discussion and Conclusions are well written, but possible hybridization of intraspecific forms of S. strandi is based on few samples. At present, it appears that the results obtained are very preliminary, raising the question of whether it is necessary to publish such interim results or whether it would be preferable to examine additional individuals and then prepare a more robust publication.

Response 15: Of course, results presented in our study are preliminary, so we present it as Comminication. Communication provides for the publication of preliminary, but new important data. More comprehensive research of S. strandi should be based on several tasks, which need more material: study of distribution of intraspecific forms, their hybridization, genetic and morphological variability etc. However, rarity of this species pushes the prospect of such research into the distant future, so we would like to publish new results of our work in the present state.”

Thank you for your comment, in generally, I can accept and support your position; however at the end of discussion and in conclusions it would be helpful if authors could change one-two sentences more highlighting the limitations of the study; also it might be helpful if authors suggest future geographical locations where samples can be taken for resolving hybridization zones, boundaries of intraspecific forms.
